# Prediction of SPT-07A Pharmacokinetics in Rats, Dogs, and Humans Using a Physiologically-Based Pharmacokinetic Model and In Vitro Data

**DOI:** 10.3390/pharmaceutics16121596

**Published:** 2024-12-15

**Authors:** Xiaoqiang Zhu, Weimin Kong, Zehua Wang, Xiaodong Liu, Li Liu

**Affiliations:** 1Center of Drug Metabolism and Pharmacokinetics, China Pharmaceutical University, Nanjing 210009, China; 3122014098@stu.cpu.edu.cn (X.Z.); zehuawang_cpu@163.com (Z.W.); 2School of Pharmacy, Bengbu Medical University, Bengbu 233030, China; wmkong@bbmu.edu.cn

**Keywords:** SPT-07A, ischemic stroke, physiologically-based pharmacokinetic model, interspecies allometric scaling, unbound fraction in plasma, UDP-glucuronosyltransferases, cytochrome P450

## Abstract

**Background/Objectives:** SPT-07A, a D-borneol, is currently being developed in China for the treatment of ischemic stroke. We aimed to create a whole-body physiologically-based pharmacokinetic (PBPK) model to predict the pharmacokinetics of SPT-07A in rats, dogs, and humans. **Methods:** The in vitro metabolism of SPT-07A was studied using hepatic, renal, and intestinal microsomes. The pharmacokinetics of SPT-07A in rats were simulated using the developed PBPK model and in vitro data. Following validation using pharmacokinetic data in rats, the developed PBPK model was scaled up to dogs and humans. **Results:** Data from hepatic microsomes revealed that SPT-07A was primarily metabolized by UDP-glucuronosyltransferase (UGTs). Glucuronidation of SPT-07A also occurred in the kidney and intestine. The in vitro to in vivo extrapolation analysis showed that hepatic clearance of SPT-07A in rats, dogs, and humans accounted for 62.2%, 87.3%, and 76.5% of the total clearance, respectively. The renal clearance of SPT-07A in rats, dogs, and humans accounted for 32.6%, 12.7%, and 23.1% of the total clearance, respectively. Almost all of the observed concentrations of SPT-07A following single or multi-dose to rats, dogs, and humans were within the 5th–95th percentiles of simulations from 100 virtual subjects. Sensitivity analysis showed that hepatic metabolic velocity, renal metabolic velocity, and hepatic blood flow remarkably affected the exposure to SPT-07A in humans. Dedrick plots were also used to predict the pharmacokinetics of SPT-07A in humans. Prediction accuracy using the PBPK model is superior to that of Dedrick plots. **Conclusions:** We elucidate UGT-mediated SPT-07A metabolism in the liver, kidney, and intestine of rats, dogs, and humans. The pharmacokinetics of SPT-07A were successfully simulated using the developed PBPK model.

## 1. Introduction

SPT-07A belongs to D-borneol extracted from traditional Chinese medicine natural borneol [1]. Accumulating evidence has demonstrated that SPT-07A exhibits beneficial effects on patients with the acute stage of ischemic stroke, including improvement of cerebral blood flow [2], reduction in cerebral infarction rate [3], suppression of Ca^2+^ overload, and protection against ROS damage [4]. In the subacute stage of ischemic stroke, SPT-07A was reported to protect the blood–brain barrier and hinder nerve cell death [5]. In the late stage of ischemic stroke, SPT-07A may promote neurogenesis and stimulate angiogenesis [6].

SPT-07A is currently being developed in China for the treatment of ischemic stroke. It was reported that following intravenous injection, SPT-07A is mainly excreted in human urine in its glucuronide, accounting for 84.69% of the administered dose [7]. The systemic clearance (CL) of SPT-07A in humans is 1942 mL/min, which is higher than the normal hepatic blood flow (1450 mL/min) [8], inferring the existence of SPT-07A extrahepatic metabolism. In addition to the liver, UGTs are also highly expressed in the kidney and intestine [9,10,11,12], contributing to the extrahepatic metabolism of SPT-07A, which needs further confirmation.

For a new candidate prior to its clinical phase, it is crucial and meaningful to gain a prediction of its pharmacokinetic profile in humans based on in vitro and animal studies. Pharmacokinetics of SPT-07A in healthy volunteers have been documented, which gives a valuable resource for validating pharmacokinetics in human from preclinical pharmacokinetic data. Physiologically-based pharmacokinetic (PBPK) model and allometric scaling are the two main approaches to predict human PK from non-clinical settings [13]. PBPK modeling is a physiologically relevant approach that integrates drug-specific and system parameters to generate pharmacokinetic predictions for targeted populations [14,15]. Allometric scaling is the empirical approach for pharmacokinetic extrapolation among interspecies, which mainly depends on animal data [16]. However, the PBPK model, a mechanistic model for predicting the pharmacokinetics of a drug, can be accomplished only from in vitro and in silico data [17]. At the same time, the PBPK model can not only estimate pharmacokinetic parameters and predict plasma and tissue concentration profiles but also gain valuable insights into compound properties, prioritize compounds for animal studies, and reduce animal trials [18,19].

The aims of the study work were: (1) to investigate the pharmacokinetics of SPT-07A in rats and dogs as well as the metabolism of SPT-07A in hepatic, intestinal, and renal microsomes of rats, dogs, and humans; (2) to identify enzymes mediating SPT-07A; (3) to develop PBPK model predicting pharmacokinetic behavior of SPT-07A using in vitro data and in silico data. Plasma concentration profiles of SPT-07A in humans were also predicted using the Dedrick method and compared with those using PBPK. The results will provide a reference for the development of new candidates.

## 2. Materials and Methods

### 2.1. Chemicals and Reagents

The analytical standard of SPT-07A was supplied by Suzhou Pharmavan Co., Ltd. (Suzhou, China). Camphorquinone, β-estradiol, zidovudine, and Uridine 5′-diphosphoglucuronic acid (UDPGA) were purchased from Sigma-Aldrich (Saint Louis, MO, USA). Alamethicin was provided by Shanghai Aladdin Biochemical Technology Co., Ltd. (Shanghai, China). NADPH tetrasodium salt was from Roche (Mannheim, Germany). Bovine serum albumin (BSA) and magnesium chloride hexahydrate (MgCl_2_·6H_2_O) were obtained from Sinopharm Chemical Reagent Co., (Shanghai, China). Mixed-gender rat liver microsomes (RLMs), dog liver microsomes (DLMs), human liver microsomes (HLMs), rat intestine microsomes (RIMs), dog intestine microsomes (DIMs), human intestine microsomes (HIMs), and human kidney microsomes (HKMs) were obtained from Xenotech (Kansas City, KS, USA). Rat kidney microsomes (RKMs) and dog kidney microsomes (DKMs) were acquired from IPHASE (Beijing, China). Recombinant UGT enzymes were purchased from Xenotech (Kansas City, KS, USA). All the other reagents were of analytical grade and commercially available.

### 2.2. In Vitro Pharmacokinetic Studies

#### 2.2.1. Blood to Plasma Partition Ratio

The blood-to-plasma concentration ratio (R_b_) was determined in triplicate per concentration (100, 300, and 900 ng/mL) for rats, dogs, and humans. The spiked blood samples were incubated on an orbital shaker for 1 h at 37 °C and then centrifuged at room temperature. The levels of SPT-07A in plasma were analyzed using the LC-MS/MS method and R_b_ was calculated.

#### 2.2.2. Plasma Protein Binding

The unbound fraction in plasma (f_u,p_) of SPT-07A was determined by dialysis using the Rapid Equilibrium Dialysis device system (RED, Thermo Fischer Scientific, Tournai, Belgium) at the concentrations of 75, 150, 300, and 600 ng/mL for all species. Aliquots of the spiked plasma were placed into the donor chamber, and 100 mM *phosphate-buffer saline* (PBS, pH 7.4) was added to the receiver chamber. The plate was covered and incubated at 37 °C at 300 rpm on an orbital shaker for 5 h. The drug concentrations in the donor and receiver chamber were measured using the LC-MS/MS method and f_u,p_ was calculated. All incubations were performed in triplicate.

#### 2.2.3. CYP-Mediated Metabolism of SPT-07A in Hepatic Microsomes

A 1.3 mL incubation system comprising RLMs (2 mg/mL), DLMs (2 mg/mL) or HLMs (2 mg/mL), SPT-07A (500 ng/mL), and NADPH (1.0 mM) was used for the experiment. Each component was formulated with 100 mM PBS (pH 7.4). Following a 5 min prewarming (37 °C), the buffer-microsome-substrate mixtures were supplemented with NADPH to commence the reaction. Two hundred μL of incubated aliquots were sampled at the designed times and the levels of SPT-07A were measured using LC-MS/MS. All incubations were performed in triplicate. The levels of SPT-07A (C) in the reaction system versus time were fitted using $ln(C)={ln(C}_{0})-kt$ on GraphPad Prism 8.0 software. The intrinsic clearance ($CL_{int,u}$) was estimated using Equation (1) [20].
(1)$$CL_{int,u}=(\frac{k}{Mic})/f_{u,mic}$$
where *k* is the negative slope of the natural log of the average concentration versus time, *Mic* is the concentration of microsome protein in the incubation system, $f_{u,mic}$ is the unbound fraction in the incubation system.

#### 2.2.4. UGT-Mediated Metabolism of SPT-07A in Hepatic, Renal, and Intestinal Microsomes

The incubation system (1.3 mL) consisted of 50 mM Tris-HCl (pH = 7.5) buffer with 1% BSA, alamethicin (50 μg/mL), MgCl_2_ (10 mM), UDPGA (1.0 mM),SPT-07A (2 μM) and RLMs (0.05 mg/mL), DLMs (0.01 mg/mL) or HLMs (0.1 mg/mL). The microsomal protein was activated utilizing alamethicin on ice for 15 min, followed by a 5 min prewarming process at 37 °C. Glucuronidation reaction was commenced with UDPGA [21,22]. Two hundred μL of incubated aliquots were sampled at the designed times and levels of SPT-07A were measured using LC-MS/MS. *CL_int,u_* was estimated as described above.

UGT-mediated metabolism of SPT-07A in renal and intestinal microsomes was also operated as described above. The levels of RKMs, DKMs, or HKMs were set to be 0.05, 0.1, and 0.1 mg/mL, respectively. The levels of RIMs, DIMs, and HIMs were all set to be 0.5 mg/mL. All incubations were performed in triplicate.

#### 2.2.5. Determining the Unbound Fraction of SPT-07A in CYP and UGT Incubation Systems

The unbound fraction ($f_{u,mic}$) in rat, dog, and human hepatic microsomes and rhUGTs was determined using 2 µM SPT-07A by the equilibrium dialysis method [23]. For the CYP incubation system, the protein concentration of RLMs, DLMs, and HLMs was set to be 2 mg/mL. For the UGT incubation system, the protein concentrations of RLMs, DLMs, HLMs, and rhUGTs were set to be 0.05, 0.01, 0.1, and 0.25 mg/mL, respectively. The plate was covered and incubated at 37 °C at 300 rpm on an orbital shaker for 5 hrs. SPT-07A concentration in the donor and receiver chamber was measured using the LC-MS/MS method and $f_{u,mic}$ were calculated. All incubations were performed in triplicate.

#### 2.2.6. Identification of rhUGTs Involved in Glucuronidation of SPT-07A

SPT-07A was incubated in the presence of 7 rhUGTs (1A1, 1A3, 1A4, 1A6, 1A9, 2B7, and 2B15) to assess which isoforms contribute to glucuronide formation. Each individual rhUGT (0.25 mg/mL) was incubated with 10 mM MgCl_2_, 1% BSA, 1 mM UDPGA, and 2 μM SPT-07A for 60 min. The levels of SPT-07A in the reaction system were measured as described above. All incubations were performed in triplicate.

#### 2.2.7. Enzyme Kinetics of SPT-07A in rhUGTs

Glucuronidation of SPT-07A was measured via clearance experiments by substrate depletion with SPT-07A, UGT1A1, and UGT2B7. The rhUGT (0.25 mg/mL) was incubated with 10 mM MgCl_2_, 1% BSA, 1 mM UDPGA, and 2 μM SPT-07A for 0, 10, 20, 30, 45, and 60 min. The levels of SPT-07A in the reaction system were measured as described above. All incubations were performed in triplicate.

#### 2.2.8. Contributions of UGT1A1 and UGT2B7 to Glucuronidation of SPT-07A in HLMs and HKMs

The percentage contributions of UGT1A1 and UGT2B7 to the glucuronidation of SPT-07A in HLMs and HKMs were estimated using the relative activity factor (RAF) method. CL_UGTs_ in HLMs and HKMs via the specific UGT isoform were estimated using Equation (2).
(2)$${CL}_{int,u}={RAF}_{UGT,i}\times {rhCL}_{int,u}$$

RAF_UGT1A1_ and RAF_UGT2B7_ were measured using the ratio of CL of β-estradiol in microsomes to that in recombinant UGT1A1, the ratio of CL of zidovudine in microsomes to that in recombinant UGT2B7, respectively. β-estradiol (1 μM) and zidovudine (1 μM) were incubated with HLMs and HKMs (levels of protein for β-estradiol: 0.1 mg/mL, 0.2 mg/mL; levels of protein for zidovudine: 0.05 mg/mL and 0.1 mg/mL) or corresponding recombinant UGT isoforms (levels of protein for specific UGT1A1 and UGT2B7 isoforms: 0.25 mg/mL) for 0, 10, 20, 30, and 45 min, respectively. β-estradiol and zidovudine in incubation mixtures were measured using LC-MS/MS methods described previously with modification [24,25].

The relative contribution of each UGT enzyme to the glucuronidation of SPT-07A was estimated by comparing each individual UGT intrinsic clearance (${CL}_{int,u}$) to intrinsic clearance in microsomes [26,27].
(3)$$f_{m}\left(\%\right)=\frac{{CL}_{int,u}}{{CL}_{int,u,mic}\;}$$

#### 2.2.9. In Vitro to In Vivo Extrapolation

The in vitro unbound intrinsic clearance (${CL}_{int,u}$) was extrapolated to the whole body level ($CL_{int,in\;vivo,u}$) via organ scaling factors according to Equation (4) [28].
(4)$$CL_{int,in\;vivo,u}=CL_{int,u}\times MPPGL\;or\;MPPGK\;or\;MPPGI\times OW$$
where *MPPGL*, *MPPGK*, and *MPPGI* represent the microsomal protein of the liver, kidney, and intestine (mg/g tissues), respectively. *OW* refers to organ weights (g/kg).

It was assumed that the disposition of SPT-07A in tissues is illustrated by the well-stirred model. The clearance of SPT-07A in tissues is estimated using Equation (5) [29].
(5)$${CL}_{b}=Q\times \frac{f_{u,b}\times {CL}_{int,in\;vivo,u}}{Q+f_{u,b}\times {CL}_{int,in\;vivo,u}}$$
where *Q* represents organ blood flow and *f_u,b_* represents the free fraction SPT-07A in blood ($f_{u,b}=f_{u,p}/R_{b}$). The contribution of each organ (as represented by hepatic, renal, and intestinal metabolism) to the total blood clearance of SPT-07A ($f_{CL,i}$) was calculated as follows.
(6)$$f_{CL,i}=\frac{{CL}_{b,i}}{{CL}_{b,tot}}$$
where ${CL}_{b,i}$ is the blood clearance for hepatic, renal, and intestinal metabolism, and ${CL}_{b,tot}$ is the sum of blood clearance for three organs. In vitro metabolism of mapping was carried out with GraphPad Prism Programs (version 8.0.1, GraphPad Software, San Diego, CA, USA).

### 2.3. In Vivo Pharmacokinetic Studies

Sprague Dawley (SD) rats were purchased from Shanghai Sippe-Bk Lab Animal Co., Ltd. (Shanghai, China). Beagle dogs were purchased from Shanghai Jiagan Biotechnology Co., Ltd. (Shanghai, China). Animals were housed for one week in a temperature- and humidity-controlled room with a 12 h light/dark cycle with free access to food and water. The experimental protocol and procedures were approved by China Pharmaceutical University Institutional Animal Care and Use Committee (no. CPU-PCPK-15030016).

#### 2.3.1. Plasma Concentration—Time Profiles in Rats

One hundred and eighty SD rats received single four doses of SPT-07A (0.5, 1, 2 mg/kg). Another batch of sixty rats was administrated by a multiple four dose (1 mg/kg, qd) for 7 days. Blood samples were collected into heparinized tubes under light ether anesthesia via the femoral artery at 5, 10, 30, 45, 60, 90, 120, 150, 180, and 240 min after administration. The plasma samples were obtained by centrifuging the blood samples at 3000 rpm for 10 min and stored at −20 °C until LC-MS/MS analysis.

#### 2.3.2. Tissue Concentration—Time Profiles in Rats

Eighteen SD rats intravenously received SPT-07A (2 mg/kg). Six rats were sacrificed under light ether anesthesia at 5, 30, and 90 min after administration. The plasma, heart, liver, spleen, stomach, brain, intestine, kidney, muscle, skin, adipose, and lung were quickly collected, homogenized, and stored at −20 °C until LC-MS/MS analysis.

#### 2.3.3. Plasma Concentration—Time Curve in Dogs

Six Beagle dogs were conducted after a four bolus was administered at single doses (0.25, 0.5, 1 mg/kg) according to a three-period cross-over design with a 3~5-day washout period. Following a 5-day washout, the dogs also received an intravenously multiple dose (1 mg/kg, qd) for 7 days. The blood samples were collected from the forearm vein of each dog into heparinized tubes at 5, 10, 20, 30, 45, 60, 90, 120, 180, 240, and 300 min after administration. The blood samples were centrifuged at 3000 rpm for 10 min to obtain the plasma samples, and stored at −20 °C until LC-MS/MS analysis.

#### 2.3.4. Clinical Data Collection

Pharmacokinetic data of SPT-07A in humans was cited from the literature [7]. Thirty-six healthy volunteers were evenly divided into three dose cohorts (10 mg, 20 mg, and 40 mg). SPT-07A was administered via 1 h intravenous infusion. The single dose was given on day 0. After 48 h, the multiple-dose started and maintained for 7 days with an interval of 12 h. The last dose was administered on the morning of day 9. The venous blood was collected before and after the first and the last dose of study treatment for the measurement of plasma concentrations using a validated LC-MS/MS method.

### 2.4. LC-MS/MS Conditions for Plasma and Metabolic Enzymes Samples Analysis

The drug concentration in plasma and metabolic enzyme samples were analyzed using a validated LC-MS/MS method. In this bioanalytical method, the samples were extracted by solid-phase. The aliquots of samples were added to the 96-well elution plate (Oasis HLB 96-well Plate 30 μm, 10 mg) to remove proteins and phospholipids. After elution in a 96-well plate, 5 µL of the eluate was injected directly into the LC-MS/MS system. Analytes were separated using an ACQUITY UPLC BEH C18 chromatographic column (1.7 μm, 2.1 mm × 50 mm, Waters, Milford, MA, USA). The mass spectrometer was operated in the positive electrospray ionization (ESI^+^) mode, and the detection of the ions was performed in the multiple reaction monitoring (MRM) mode, monitoring the transition of *m*/*z* 137.1 to *m*/*z* 81.0 for SPT-07A and *m*/*z* 167.2 to *m*/*z* 83.2 for internal standard (IS). The lower limit of quantification was 1.95 ng/mL for plasma studies and 0.1 μM for metabolic enzyme studies.

### 2.5. PBPK Modeling of SPT-07A

#### 2.5.1. PBPK Model Development

A PBPK model (Figure 1) was constructed to describe the pharmacokinetic profiles of SPT-07A in rats, dogs, and humans. The essential structure of the model consisted of the lung, heart, brain, muscle, skin, adipose, kidney, liver, spleen, stomach, intestine, arterial, venous blood, and the rest of the body. All tissues except for the adipose are represented by a perfusion-rate limited model. SPT-07A was eliminated in the liver, kidney, and intestine. The differential equations are described as follows [30,31]:

For non-elimination tissue (except for the adipose) compartments (*t*),
(7)VtⅆCtdt=Qt×Cart−CtKt:pl∕Rb

For arterial blood compartment (*art*),
(8)VartⅆCartdt=Qtotal×ClunKlun:pl/Rb−Cart

For venous blood compartment (*ven*),
(9)VvenⅆCvendt=∑tQt×CtKt:pl∕Rb−Qtotal×Cven

For lung compartment (*lun*),
(10)VlunⅆClundt=Qtotal×Cven−ClunKlun:pl/Rb
where Ct, Vt, Qt, and Kt:pl are the concentration of SPT-07A, the volume, the blood flow rate and tissue-to-plasma concentration ratio of corresponding tissue. Rb refers to the blood-to-plasma concentration ratio. Qtotal represents the cardiac output.

Tissue distribution studies in rats have showed that high accumulation of SPT-07A occurs in the adipose. A permeability-limited model was introduced to illustrate the disposition of SPT-07A in the adipose [32,33].
(11)$$V_{1}\frac{dC_{1}}{dt}=Q_{adi}\left(C_{art}-C_{1}\right)-PS\left(C_{1}-\frac{C_{2}}{K_{t:pl}/R_{b}}\right)$$
(12)$$V_{2}\frac{dC_{2}}{dt}=PS\left(C_{1}-\frac{C_{2}}{K_{t:pl}/R_{b}}\right)$$
where subscripts *adi*, 1 and 2 represent the adipose tissue, vascular and extravascular compartments, respectively. The values of $V_{1}$ and $V_{2}$ are derived from an open-source PK-Sim software (version 11.2). Specifically, for rats, they are 2.66 mL and 16.34 mL; for dogs, 210 mL and 1290 mL; for humans, 1800 mL and 8200 mL. PS is the permeability-surface product, which was estimated by fitting this value to the adipose concentration data in rats. The values of PS in other species were estimated by Equation (13).
(13)$${PS}_{i}={PS}_{rat}\left(\frac{W_{i}}{W_{rat}}\right)$$
where *W_i_* was the weight of the indicated species.

For the elimination of tissue (liver, kidney, and intestine) compartments [34],
(14)VlivⅆClivdt=Qhep×Cart+Qsto×CstoKsto:pl/Rb+Qinte×CinteKinte:pl/Rb+Qspl×CsplKspl:pl/Rb−Qhep+Qsto+Qinte+Qspl×ClivKliv:pl/Rb−CLliv_int,in vivo,u×W×fu,p×ClivKliv:pl
(15)VkidⅆCkiddt=Qkid×(Cart−CkidKkid:pl/Rb)−CLkid_int,in vivo,u×W×fu,p×CkidKkid:pl
(16)VinteⅆCintedt=Qinte×(Cart−CinteKinte:pl/Rbp)−CLinte_int,in vivo,u×W×fu,p×CinteKinte:pl
where subscripts *liv*, *kid*, *inte*, *sto* and *spl* represent the liver, kidney, intestine, stomach and spleen, respectively. Qhep represents blood flow rate of hepatic artery. CLliv_int,in vivo,u, CLkid_int,in vivo,u and CLinte_int,in vivo,u are the unbound in vivo whole organ intrinsic clearance, respectively. W is body weight.

$K_{t:pl}$ of SPT-07A in rast, dogs, and humans were estimated according to the Schmitt method based on the tissue composition of the corresponding species [35,36]. Fraction of water, interstitial, lipid, neutral lipid, neutral phospholipids, acidic phospholipids, protein in tissues of various species and the $f_{u,p}$ of SPT-07A were incorporated into the calculation.

#### 2.5.2. Validation of the Developed PBPK Model

The plasma concentrations of STP-07A following intravenous dose to 100 virtual animals and humans were predicted on the Phoenix WinNonlin software (Version 8.3, Certara, USA, Inc., Princeton, NJ, USA) using the parameters listed in Table 1 and Table 2. The simulations were further compared with observations.

Two aspects were used to comprehensively evaluate the prediction performance of the PBPK model: (i) the observed pharmacokinetic profiles were within the 5th–95th percentiles of simulations based on 100 virtual subjects, (ii) the predicated pharmacokinetic parameters such as the area under the concentration–time curve (AUC) and the maximum plasma concentration (C_max_) were within the 0.5–2.0 folds of observations.

**Table 1 pharmaceutics-16-01596-t001:** Physiological parameters and drug parameters for simulation of SPT-07A in the PBPK model.

	Rat (0.25 kg)			Dog (8.5 kg)			Human (70 kg)		
	Volume (mL)	Blood Flow (mL/min)	K_t:pl_	Volume (mL)	Blood Flow (mL/min)	K_t:pl_	Volume (mL)	Blood Flow (mL/min)	K_t:pl_
Lung	1.25	83.90	0.59	85	1120	0.62	1170	5600	0.76
Heart	0.83	4.07	1.13	43	43.3	0.76	310	240	3.62
Brain	1.43	1.66	1.46	50	145	1.44	1450	700	4.16
Muscle	117.50 ^a^	8.23 ^b^	0.90	4250	270 ^a^	0.81	35,000	750	1.19
Adipose	19.00	5.82	1.92	1500	50	1.72	10,000	260	2.84
Skin	47.50	4.82	7.20	774 ^c^	71.5 ^a^	6.46	7800	300	8.18
Kidney	1.83	11.71	1.51	40 ^b^	170	0.79	280	1240	2.22
Spleen	0.50	1.66	0.98	22	13.33	0.49	190	80	1.27
Stomach	1.10	1.13	1.00	24	10	1.00	160	38.33	1.00
Liver	9.15	12.30	1.53	213	323.33	0.81	1690	1518.33	2.06
Vein	13.60	–	–	284	–	–	3470	–	–
Artery	6.80	–	–	141	–	–	1730	–	–
Liver-art	–	1.99	–	–	45	–	–	300	–
intestine	10.01	7.52	1.00	203	255	1.00	1650	1100	1.00
Rest of body	19.50	35.29	1.00	871	48.33	1.00	5100	592	1.00
MPPGL	44.8			63.6			48.8		
MPPGK	17.9			44.0			17.8		
MPPGI	9.7			6.5			0.54		
R_b_	1.10			0.92			0.92		
f_u,p_	28.70%			25.77%			32.64%		

^a^ values were the average value from PK-Sim software (version 11.2) and [37]; ^b^ values were from an open-source PK-Sim software (version 11.2); ^c^ value was from [38]. The rest of the physiological parameters were from [37].

**Table 2 pharmaceutics-16-01596-t002:** Enzyme kinetic parameters for CYP and UGT-mediated SPT-07A metabolism in hepatic, renal, and intestinal microsomes of rats, dogs, and humans.

Species	Microsomes	Enzymes	*CL_int,u_*	*CL_int,_* _in vivo*,u*_	Total *CL_int,_*_in vivo*,u*_	*CL_in.vivo_*	Contribution
			μL/min/mg Protein	mL/min/kg	mL/min/kg	mL/min/kg	(%)
Rat	RLMs	UGT	2060	3380	3390	46.6	62.2
		CYP	5.14	8.43			
	RKMs	UGT	1500	196	196	24.4	32.6
	RIMs	UGT	79.1	17.2	17.2	3.90	5.2
Dog	DLMs	UGT	12,200	25,500	25,500	37.8	87.3
		CYP	13.6	28.4			
	DKMs	UGT	112	27.1	27.1	5.50	12.7
	DIMs	UGT	–	–	–	–	
Human	HLMs	UGT	745	934	938	20.4	76.5
		CYP	3.35	4.20			
	HKMs	UGT	339	26.5	26.5	6.14	23.1
	HIMs	UGT	39.8	0.309	0.309	0.109	0.4

MPPGL in rats and humans from [39], MPPGL in dogs from [40], MPPGK in rats from [41], MPPGK in dogs from [40], MPPGK in humans was the average value from [40,42], MPPGI in rats from [43], MPPGI in dogs from [40], MPPGI in humans from [44]. 

#### 2.5.3. Parameter Sensitivity Analysis

To identify the quantitative impact of key factors that influence plasma C-T profiles of SPT-07A, we performed a parametric sensitivity analysis on the developed PBPK model. SPT-07A is mainly eliminated by liver and kidney metabolism. In addition, SPT-07A possesses a high adipose-to-plasma concentration ratio. Therefore, the sensitivity analysis was conducted to evaluate the impacts of alterations in the metabolic velocity of SPT-07A in the liver and kidney, $f_{u,p}$, $K_{adipose:plasma}$, hepatic blood flow rate, and kidney blood flow rate on pharmacokinetic behavior in humans. The variations were set to be 10-fold for metabolic velocity, 2-fold for $f_{u,p}$, 2-fold for $K_{adipose:plasma}$ and 2-fold for physiological parameters.

#### 2.5.4. Interspecies Scaling by Dedrick Method

We tried to predict the pharmacokinetics of SPT-07A in humans by the Dedrick method. In brief, assuming that $V_{ss}$ is proportional to $W^{\beta 1}$ and CL is proportional to $W^{\beta 2}$. $V_{ss}$ and CL of SPT-07A across rats and dogs are expressed as follows:(17)$$V_{ss}=\alpha_{1}W^{\beta 1}\;\mathrm{and}\;CL=\alpha_{2}W^{\beta 2}$$
where *W* is the body weight of the corresponding species, *α* and *β* are the coefficient and exponent, respectively. The plasma C-T data of SPT-07A from different species were graphed using a Dedrick plot, in which *X*-axis was the physiological time $t^{\prime }$ ($t^{\prime }=t/W^{\beta 1-\beta 2}$) and *Y*-axis was normalized concentration $C^{\prime }$ $\left[C^{\prime }=C/({D/W}^{\beta 1})\right]$, where *D* represents the dose [45]. The normalized concentration-physiological time data were fitted to a two-compartment model. The plasma concentration-time profiles and pharmacokinetic parameters (k_10_, k_12_, k_21_, and V_1_) of SPT-07A in a 70 kg human were then obtained via reverse transformation from the Dedrick plot. Then, plasma C-T profiles and PK parameters in humans following intravenous infusion were obtained. Single-species (SSS_rat_, SSS_dog_, where exponents for β_1_ and β_2_ were 1.00 and 0.75, respectively) and two-species (Elementary Dedrick Plot TS_rat-dog_, where the exponent for β_1_ was 1.00 and β_2_ was derived from interspecies scaling. Complex Dedrick Plot TS_rat-dog_, where the exponents for both β_1_ and β_2_ were obtained from interspecies scaling) were used to predict the pharmacokinetics of SPT-07A [46]. The predicted C-T profiles and the estimated pharmacokinetic parameters in humans were compared with those from the PBPK model and clinic observations.

#### 2.5.5. Prediction Accuracies of Simulating SPT-07A Concentrations in Human Plasma Using Five Methods

Five methods (PBPK model, SSS_rat_, SSS_dog_, Elementary Dedrick Plot TS_rat-dog,_ and Complex Dedrick Plot TS_rat-dog_) were used to simulate the plasma concentration of SPT-07A in human [47]. The prediction accuracies were evaluated using geometric mean-fold error (*GMFE*) and the root mean square error (*RMSE*) [48]:(18)$$GMFE=10^{\frac{\sum log(C_{pre}/C_{obs})}{n}}$$
(19)$$RMSE=\sqrt{\frac{\sum {\left(C_{pre}-C_{obs}\right)}^{2}}{n}}$$
where *C_pre_* and *C_obs_* are the predicted concentrations and the observed concentrations, respectively. *n* is the number of samples.

## 3. Results

### 3.1. In Vitro PK Studies

#### 3.1.1. Blood-Plasma Partition Ratios of SPT-07A and Plasma Protein Binding Rates of SPT-07A in Rats, Dogs, and Humans

The $R_{b}$ of SPT-07A for rats at 100, 300, and 900 ng/mL were 1.11, 1.09, and 1.11, respectively. The $R_{b}$ values of SPT-07A for dogs at 100, 300, and 900 ng/mL were 0.92, 0.92, and 0.91, respectively. The $R_{b}$ values of SPT-07A for humans at 100, 300, and 900 ng/mL were 0.90, 0.93, and 0.92, respectively. The average $R_{b}$ values of SPT-07A at the tested three concentrations in rats, dogs, and humans were 1.10, 0.92, and 0.92, respectively.

Rat plasma protein binding rates at 75, 150, 300, and 600 ng/mL were 68.82 ± 6.60, 72.40 ± 5.38, 73.19 ± 0.95, 70.79 ± 2.65%, respectively. Dog plasma protein binding rates at 75, 150, 300, and 600 ng/mL were 73.78 ± 0.64, 74.31 ± 1.23, 75.50 ± 0.84, 73.34 ± 0.80%, respectively. Human plasma protein binding rates at 75, 150, 300, and 600 ng/mL were 63.35 ± 1.28, 69.30 ± 2.16, 68.04 ± 0.66, 68.76 ± 0.76%, respectively. The average protein binding rates of SPT-07A in rat, dog, and human plasma were 71.30%, 74.23%, and 67.36%, respectively. The $f_{u,p}$ of SPT-07A in rat, dog, and human plasma were estimated to be 28.70%, 25.77%, and 32.64%, respectively.

#### 3.1.2. Enzyme Kinetics and the Clearance of SPT-07A CYP and UGT Metabolism in Microsomes

The $f_{u,mic}$ values of SPT-07A in RLMs, DLMs, and HLMs in the CYP incubation system were measured to be 0.95, 0.95, and 0.98, respectively. No sspecies differencewas observed. However, the $f_{u,mic}$ values of SPT-07A in RLMs, DLMs, and HLMs in the UGT incubation system in the presence of 1% BSA were decreased to 0.55, 0.61, and 0.56, respectively. The $f_{u,mic}$ values of SPT-07A in the UGT1A1- and UGT2B7- incubation system were measured to be 0.61 and 0.60, respectively. The $f_{u,mic}$ values of SPT-07A among the five UGT incubation systems were similar, indicating that the decreases in $f_{u,mic}$ of SPT-07A were mainly attributed to 1% BSA.

The CYP enzyme kinetic and intrinsic clearance values were investigated in the presence of CYP cofactors (Table 2; Figure 2A). As measured by substrate depletion, the CYP-mediated $CL_{int,u}$ values in RLMs, DLMs, and HLMs were 5.14, 13.6, and 3.35 μL/min/mg protein, respectively. No CYP-mediated metabolism of SPT-07A occurred in the renal and intestinal microsomes of rats, dogs, and humans. The CYP-mediated ${CL}_{int,in\;vivo,u}$ values in the liver of rats, dogs, and humans were scaled to be 8.43, 28.4, and 4.20 mL/min/kg body weight, respectively.

UGT-mediated glucuronidation of SPT-07A in microsomes of the liver, kidney, and intestine in rats, dogs, and humans was measured (Table 2; Figure 2B–D). The UGT-mediated $CL_{int,u}$ values in RLMs, RKMs, and RIMs were measured to be 2060, 1500, and 79.1 μL/min/mg protein, respectively. And the scaled ${CL}_{int,in\;vivo,u}$ values in the liver, kidney, and intestine of rats were 3380, 196, and 17.2 mL/min/kg body weight, respectively.

The estimated UGT-mediated $CL_{int,u}$ values in DLMs and DKMs were 12,200 and 112 μL/min/mg protein, respectively. No glucuronidation of SPT-07A was observed in DIMs. The scaled ${CL}_{int,in\;vivo,u}$ values in the liver and kidney of dogs were 25,500 and 27.1 mL/min/kg body weight, respectively.

Glucuronidation of SPT-07A occurred in HLMs, HKMs, and HIMs. The UGT-mediated $CL_{int,u}$ values were measured to be 745, 339, and 39.8 μL/min/mg protein, respectively. The scaled ${CL}_{int,in\;vivo,u}$ values in the liver, kidney, and intestine of humans were 934, 26.5, and 0.309 mL/min/kg body weight, respectively.

#### 3.1.3. The Contributions of UGT-Mediated Metabolism of SPT-07A in the Liver, Intestine, and Kidney to Systemic Clearance of SPT-07A in Rats, Dogs, and Humans

Clearance of SPT-07A in the liver, intestine, and kidney were estimated using the well-stirred model. The estimated clearance values of SPT-07A in the liver, intestine, and kidney of rats were 46.6, 3.90, and 24.4 mL/min/kg, respectively. The systemic clearance of SPT-7A in rats was calculated to be 75.0 mL/min/kg. Contributions of the UGT-mediated metabolism of SPT-07A in the liver, intestine, and kidney of rats to systematic clearance of SPT-07A were 62.2%, 5.2%, and 32.6%.

The estimated clearance values of SPT-07A in the liver and kidney of the dog were 37.8 and 5.50 mL/min/kg, respectively. The systematic clearance of SPT-7A in dogs was calculated to be 43.3 mL/min/kg. Contributions of UGT-mediated metabolism of SPT-07A in the liver and kidney of dogs to systemic clearance of SPT-07A were 87.3% and 12.7%.

The estimated clearance values of SPT-07A in the liver, intestine, and kidney of humans were 20.4, 0.109, and 6.14 mL/min/kg, respectively. The systematic clearance of SPT-7A in humans was calculated to be 26.6 mL/min/kg. Contributions of UGT-mediated metabolism of SPT-07A in the liver, intestine, and kidney of humans to systemic clearance of SPT-07A were 76.5%, 0.4%and 23.1%. All these results indicated that UGT-mediated metabolism of SPT-07A in the liver and kidney showed an important contribution to SPT-07A systematical clearance of SPT-07A in rats, dogs, and humans.

#### 3.1.4. In Vitro UGT Phenotyping of SPT-07A

UGTs involved in glucuronidation of SPT-07A were identified using rhUGTs. After incubating SPT-07A with recombinant human UGT1A1, UGT1A3, UGT1A4, UGT1A6, UGT1A9, UGT2B7, and UGT2B15 for 60 min, the remaining amounts of SPT-07A were 75.2%, 96.5%, 99.5%, 97.5%, 98.9%, 40.2%, and 98.4%, respectively (Figure 2E). SPT-07A was almost stable in the UGT1A3, UGT1A4, UGT1A6, UGT1A9, and UGT2B15 incubation system, indicating that UGT1A1 and UGT2B7 mainly contributed to catalyzing the glucuronidation of SPT-07A.

Glucuronidation kinetics of SPT-07A in rhUGT1A1 and rhUGT2B7 incubation systems were measured. Enzyme kinetics of SPT-07A in rhUGT1A1 and rhUGT2B7 exhibited first-order kinetics (Figure 2F). The estimated ${rhCL}_{int,u}$ values in rhUGT 1A1 and 2B7 were 31.6 and 101 μL/min/mg protein, respectively.

#### 3.1.5. The Contributions of UGT1A1 and UGT2B7 to Glucuronidation of SPT-07A in HLMs and HKMs

The contributions of UGT1A1 and UGT2B7 to the glucuronidation of SPT-07A in HLMs and HKMs were estimated using the RAF approach. The activities of UGT1A1 and UGT2B7 in HLMs and HKMs were measured using glucuronidations of β-estradiol and zidovudine (Figure 2G–H). No β-estradiol metabolism was observed in HKMs. Intrinsic clearances of β-estradiol in rhUGT1A1 and HLMs were measured to be 50.0 and 61.9 μL/min/mg protein. The RAF_UGT1A1_ in HLMs was calculated to be 1.24. The contributions of UGT1A1 to glucuronidation of SPT-07A in HLMs was only 5.3%.

Intrinsic clearances of zidovudine in rhUGT2B7, HLMs, and HKMs were measured to be 85.8, 193, and 138 μL/min/mg protein. The RAF_UGT2B7_ values in HLMs and HKMs were calculated to be 2.25 and 1.61. The contributions of UGT2B7 to glucuronidation of SPT-07A in HLMs and HKMs were estimated to be 30.5% and 47.7%. The predicted total intrinsic clearance of SPT-07A glucuronidation in HLMs and HKMs using the RAF approach were 265 and 162 μL/min/mg, respectively, which were obviously less than observations (745 μL/min/mg in HLMs and 339 μL/min/mg in HKMs, indicating that other UGTs may mediate SPT-07A glucuronidation). Moreover, differences between the activity of UGT2B7 in HLM and in rhUGT2B7 also lead to the underpredictions, which may explain the report that K_m_ of UGT2B7 mediated zidovudine glucuronidations in HLMs were 1.6 folds of that in rhUGT2B7 [49].

### 3.2. PBPK Modeling

#### 3.2.1. The Simulation of SPT-07A Pharmacokinetics in Rats Using PBPK Model

The plasma concentrations of SPT-07A following intravenous dose to 100 virtual rats were predicted using the developed PBPK model and the in vitro pharmacokinetic parameter (Table 1 and Table 2) and compared with observations. The results showed that all of the observed plasma concentrations fell within the 5th–95th percentile range of simulations (Figure 3). The ratio of the predicted AUC, CL, and t_1/2_ to the observed values were 0.95~1.22, 0.87~1.06, and 0.54~0.97, respectively (Table 3).

Distribution of SPT-07A in the heart, liver, spleen, stomach, brain, intestine, kidney, muscle, adipose, and lung of rats following intravenous 2 mg/kg to rats was also simulated. The results showed that except for the kidney of male rats, almost all of the observed concentrations fell within the 5th–95th percentile range of simulations (Figure 4A–K). The predicted exposures (AUC_0–90 min_) from the predicted profiles in the tested tissues except the kidney were within 0.5–2.0 folds of observations (Figure 4L).

#### 3.2.2. The Simulation of SPT-07A Pharmacokinetics in Dogs Using PBPK Model

Following validation in rats, the developed PBPK model was scaled up to dogs. The plasma concentrations of SPT-07A in dogs following single or multidose of SPT-07A were simulated and compared with observations (Figure 5). The results showed that all of the observed concentrations were within the 5th–95th percentile range of simulations. The estimated pharmacokinetic parameters were also with 0.5–2.0 folds of observations (Table 3).

#### 3.2.3. The Simulation of SPT-07A Pharmacokinetics in Humans Using PBPK Model

The developed PBPK model was further scaled up to humans. The plasma concentration of SPT-07A in 100 virtual humans via the single dose of 1 h intravenous infusion or multidose of 1 h intravenous infusion were simulated and compared with clinical observations (Figure 6). Almost all of the observed concentrations of SPT-07A were within the 5th–95th percentile range of simulations. The estimated pharmacokinetic parameters were with 0.5–2.0 folds of observations, except the t_1/2_ in the medium-dose group of multiple doses (Table 3). All these results indicated that the plasma concentrations of SPT-07A in humans may be successfully predicted using the developed PBPK model and in vitro metabolic parameters.

### 3.3. Sensitivity Analysis

The impact of metabolic velocity and physiological parameters of SPT-07A on the pharmacokinetics behavior in humans was investigated, and the results were presented in Figure 7 and Table 4. Reducing hepatic metabolic velocity remarkably affected the exposure of SPT-07A. When the hepatic metabolic velocity was reduced to 1/10, C_max_ and AUC_last_ ratios increased to 1.22 and 1.36, respectively. CL ratios were reduced to 0.73. However, t_1/2_ ratios changed by 8%. When the hepatic metabolic velocity was increased 10-fold, the ratios of PK parameters (C_max_, AUC_last_, CL, and t_1/2_) changed by a maximum of 4%. Renal metabolic velocity could change the exposure of SPT-07A. A 10-fold increase in renal metabolic velocity led to the changes in C_max_, AUC_last,_ and CL ratios to 0.81, 0.76, and 1.33 of the initial values, while a 1/10-fold increase in renal metabolic velocity led to the changes in C_max_, AUC_last,_ and CL ratios to 1.16, 1.24, and 0.80 of the initial values. Altering the metabolic rate of the kidney affects t_1/2_ by 5%. f_u,p_ remarkably affected the PK parameters of SPT-07A. A 2-fold increase in f_u,p_ resulted in the changes of C_max_, AUC_last_, CL, and t_1/2_ ratios to 0.91, 0.88, 1.14, and 0.98, respectively, compared to the initial values. When f_u,p_ was set to be 1/2 of the initial values, C_max_, AUC_last,_ and CL and t_1/2_ ratios were 1.10, 1.15, 0.86, and 1.03, respectively. The impact of a 2-fold alteration in K_adipose:plasma_ on the major PK parameters (C_max_, AUC_last,_ and CL) was approximately 1%. The effect on t_1/2_ was approximately 10%. The impact of the hepatic blood flow on C_max_, AUC_last,_ and CL exceeded t_1/2_. When the hepatic blood flow was doubled, C_max_, AUC_last,_ and CL ratios were 0.68, 0.60, and 1.68, respectively. When the hepatic blood flow was set to be 1/2 of the initial values, C_max_, AUC_last,_ and CL ratios were 1.34, 1.56, and 0.63, respectively. The maximum effect of a 2-fold alteration in the renal blood flow rate on the major PK parameters was 6%.

### 3.4. Interspecies Allometric Scaling by Dedrick Methods

The observed and predicted C-T profiles obtained by the PBPK and Dedrick methods are shown in Figure 8. The fold-error and prediction performances of plasma concentrations were also summarized in Table 5.

For the prediction of plasma concentration, the GMEF and RMSE values of PBPK, SSS_rat,_ and Elementary TS_rat-dog_ methods were similar, and the above three methods were superior to SSS_dog_ and Complex TS_rat-dog_ methods. For the prediction of PK parameters, the predictability of the SSS_rat_ approach was the lowest for C_max_, whereas the other four methods exhibited similar performance. The prediction of AUC was most accurately achieved using the PBPK approach. Taken together, the overall ability of the PBPK approach is superior to one- and two-species by the Dedrick approach in anticipating human PK of SPT-07A. Among the four Dedrick methods, the Elementary TS_rat-dog_ method has the best prediction ability for drug concentrations and PK parameters.

## 4. Discussion

In this study, we first systematically investigated the pharmacokinetics of SPT-07A. The SPT-07A metabolism mainly accounted for UGT-mediated glucuronidation in the liver of rats, dogs, and humans. Pharmacokinetics of SPT-07A in rats, dogs, and humans were successfully simulated using the developed PBPK model based on in vitro pharmacokinetic parameters derived from microsomes of liver, kidney, and intestine.

The microsomal enzyme kinetics studies demonstrated that the rat, dog, and human systemic clearance was expected to be mediated by UGT, while CYP-mediated metabolism was negligible. These results were consistent with the mass balance study in healthy volunteers [7]. Using the substrate elimination method in microsomal enzyme kinetics can accurately predict both UGT and CYP-mediated metabolism of SPT-07A in human [50,51].

The glucuronidation metabolic clearance of SPT-07A was tissue- and species-dependent based on microsomal enzyme kinetics experiments. For tissue dependence, the CL_int,u_ values were observed in the order of liver > kidney > intestine in rats, dogs, and humans [52]. For species dependence, The CL_int,u_ values in liver microsomes were the highest in dogs, followed by rats and then humans. In rat kidney microsomes, the CL_int,u_ was approximately five times higher than that in human kidney microsomes, while the CL_int,u_ in dog kidney microsomes was the lowest. Although we observed that SPT-07A glucuronidation was mediated by intestine microsomes, the intestine-mediated metabolism of SPT-07A in rats, dogs, and humans was very limited [53,54]. The rhUGTs enzyme kinetics indicated that SPT-07A was primarily catalyzed by UGT2B7 with minor contributions from UGT1A1. UGT2B7, a high-expression isoform of UGT in the human liver and kidney, is responsible for the glucuronidation of SPT-07A [55,56]. The protein expression levels of UGT2B7 in human liver, kidney, and intestine microsomes were 112.83, 59.77, and 6.86 pmol/mg protein of microsomes, respectively [57,58]. The CL_int,u_ of SPT-07A in the liver, kidney, and intestine were basically proportional to the expression level of UGT2B7 in corresponding microsomes. DKMs and DIMs exhibited a lack of glucuronidation activity. This may be attributed to the limited protein expression of renal UGTs and intestinal UGT2B7 in DKMs and DIMs, respectively [59,60]. The quantitative and comprehensive analyses of UGT2B7 expression levels in rat microsomes had not yet been reported. These results suggest that different protein expression levels of UGT2B7 in the liver, kidney, and intestine lead to an extensive tissue- and species-dependence in the glucuronidation of SPT-07A among dogs and humans.

The ER values (extraction ratio) in the liver were 0.95, 0.99, and 0.94 in rats, dogs, and humans, respectively. SPT-07A is almost completely metabolized by hepatic first-pass metabolism after oral administration in rats and dogs. The rats were orally administered 5 mg/kg of SPT-07A, resulting in minimal blood exposure. The oral bioavailability in humans is inferred no more than 6%. Thus, SPT-07A is not suitable for oral administration. To avoid hepatic first-pass metabolism, intravenous or sublingual administration was a good way to treat ischemic stroke with SPT-07A [61,62]. The in vitro metabolism significantly influences the choice of drug delivery routes.

The tissue distribution of SPT-07A in rats revealed sex differences only in the kidney, with higher concentrations found in the male kidney compared to the female kidney. The beta-Glu activity was stronger in male rat kidney [63], while the expression of UGT enzymes was higher in female rat kidney [64]. This may explain why the concentration of SPT-07A was obviously higher in male rat kidneys. Since the variance between the beta-Glu activity and the expression of UGT enzymes in rat kidneys was not taken into account, the established PBPK model had a low prediction of renal tissue concentration in male rats.

SPT-07A was primarily metabolized by UGT2B7. Therefore, special attention should be given to potential drug–drug interactions (DDIs) when co-administration with UGT2B7 inhibitors in patients [65,66]. According to the well-stirred model, SPT-07A exhibited high and intermediate extraction ratios in human liver and kidney, respectively. Sensitivity analysis demonstrated that hepatic metabolic velocity, renal metabolic velocity, and hepatic blood flow remarkably impacted the exposure of SPT-07A in humans. Elderly stroke patients with severe hepatic and renal impairment may experience reductions in hepatic and renal blood flow rates, albumin levels, hematocrit values [67,68], as well as protein abundance levels of UGT2B7 [69,70]. These changes could substantially alter the pharmacokinetic behavior and impact the safety and efficacy of SPT-07A. The established PBPK model serves as a valuable tool for optimizing dosing in populations with hepatic or renal impairment and evaluating DDIs during clinical studies [71].

The UGT2B7 gene (chromosome 4) is also polymorphic, containing three nonsynonymous, as well as many synonymous, intronic, and promoter SNPs. To date, a total of four alleles of UGT2B7 have been described, which may affect UGT2B7 expression and glucuronidation activity [72]. Research shows that UGT2B7 polymorphisms may be an important determinant of individual variability in the pharmacokinetics of drugs like valproic acid and tamoxifen [73,74]. The UGT2B7 polymorphisms profile in West African and Papua New Guinean populations is similar to African-Americans, but different from Asian-Americans [75]. The European and Asian populations showed higher pairwise differentiation values for UGT2B7 [72]. These studies suggest possible inter-ethnic variability in the pharmacokinetics of UGT2B7-mediated metabolic drugs. The polymorphism of UGT2B7 suggests that SPT-07A needs a population pharmacokinetic study. In the future, its pharmacokinetics should be studied among different ethnicities, if it goes into the global development phase. These pharmacokinetic studies were used to evaluate the impact of individual and ethnic variability on the effect of SPT-07A, as well as its safety and efficacy under a specific dosage and dosage regimen.

SPT-07A exhibits microsomal interspecies differences and renal-mediated extrahepatic metabolism. Allometric scaling can be applied to drugs whose elimination was either renal or liver-blood flow dependent. However, it may be misleading for some drugs when interspecies differences, extrahepatic metabolism, and low metabolic organ extraction ratio are not taken into account [76,77]. Interspecies differences in metabolism and extrahepatic metabolism can be researched by the PBPK model. This may be the reason why PBPK gave more accurate predictions of pharmacokinetic parameters and C-T profiles than the Dedrick approach [78,79].

## 5. Conclusions

In summary, renal-mediated glucuronidation was an essential pathway in the elimination of SPT-07A besides hepatic glucuronidation metabolism in rats, dogs, and humans. SPT-07A was primarily metabolized by UGT2B7 with minor contributions from UGT1A1. The PBPK model was successfully established based on the in vitro clearance in rats, dogs, and humans. These studies enabled further interpretation of SPT-07A disposition and drug–drug interaction risk assessment with coadministered drugs and could be used for optimizing dosing in populations with hepatic or renal impairment.

## Figures and Tables

**Figure 1 pharmaceutics-16-01596-f001:**
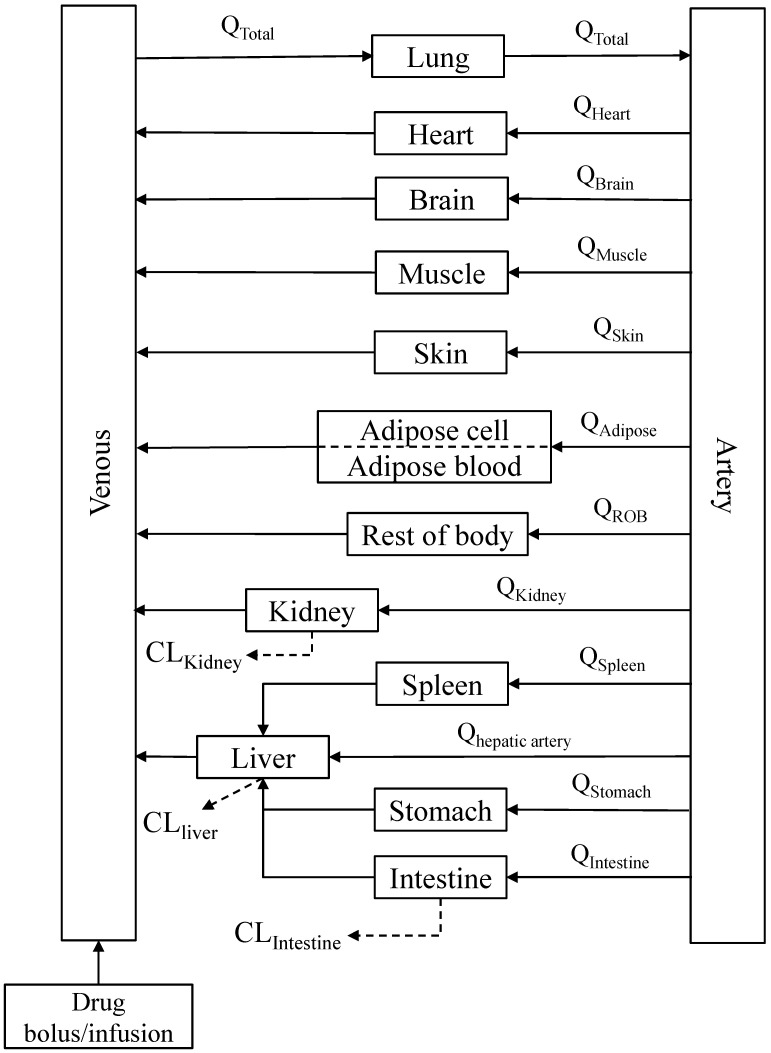
Schematic diagram of whole-body PBPK model of SPT-07A with major tissues (Arrows connecting compartments stand for the blood flows. ROB represents the rest of the body).

**Figure 2 pharmaceutics-16-01596-f002:**
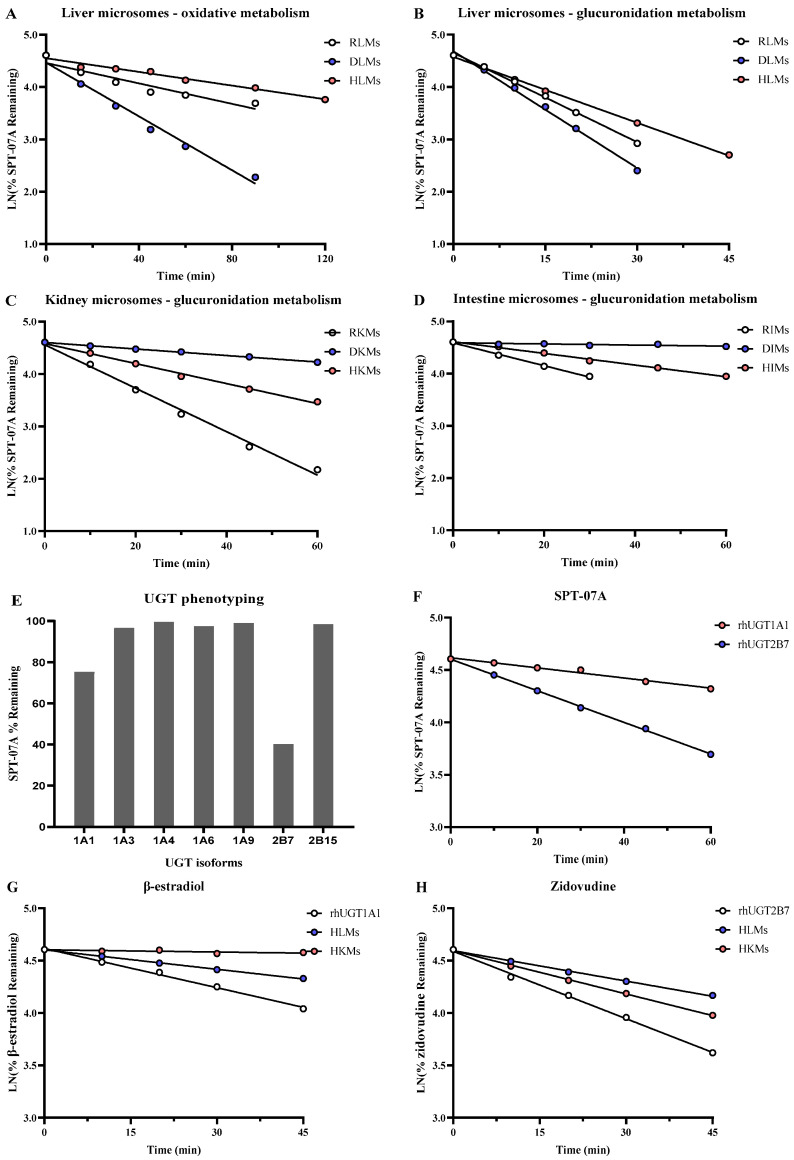
SPT-07A metabolism in CYP450 incubation system of liver microsomes (**A**). SPT-07A glucuronidation metabolism in liver microsomes (**B**), kidney microsomes (**C**), and intestine microsomes (**D**). SPT-07A glucuronidation in rhUGTs (**E**). SPT-07A glucuronidation in rhUGT1A1 and rhUGT2B7 (**F**). β-estradiol glucuronidation in HLMs, HKMs, and rhUGT1A1 (**G**). Zidovudine glucuronidation in HLMs, HKMs and rhUGT2B7 (**H**).

**Figure 3 pharmaceutics-16-01596-f003:**
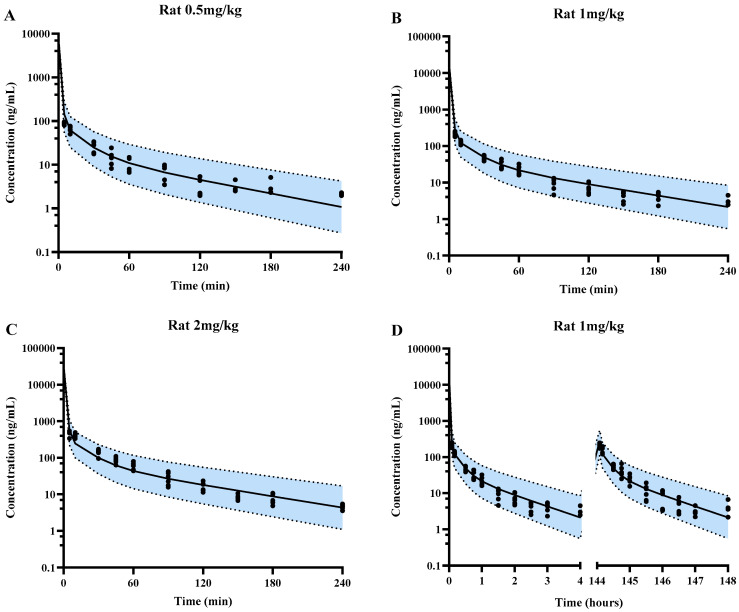
Predicted population mean (black solid lines) and observed (black solid circles) plasma concentration-time profiles of SPT-07A in rats for single-dose 0.5 mg/kg (**A**), single-dose 1 mg/kg (**B**), single-dose 2 mg/kg (**C**) and multiple-dose 1 mg/kg (**D**). Blue areas represent 90% prediction intervals (5th–95th percentile boundaries; lower and upper black dashed lines, respectively) for a virtual population (n = 100).

**Figure 4 pharmaceutics-16-01596-f004:**
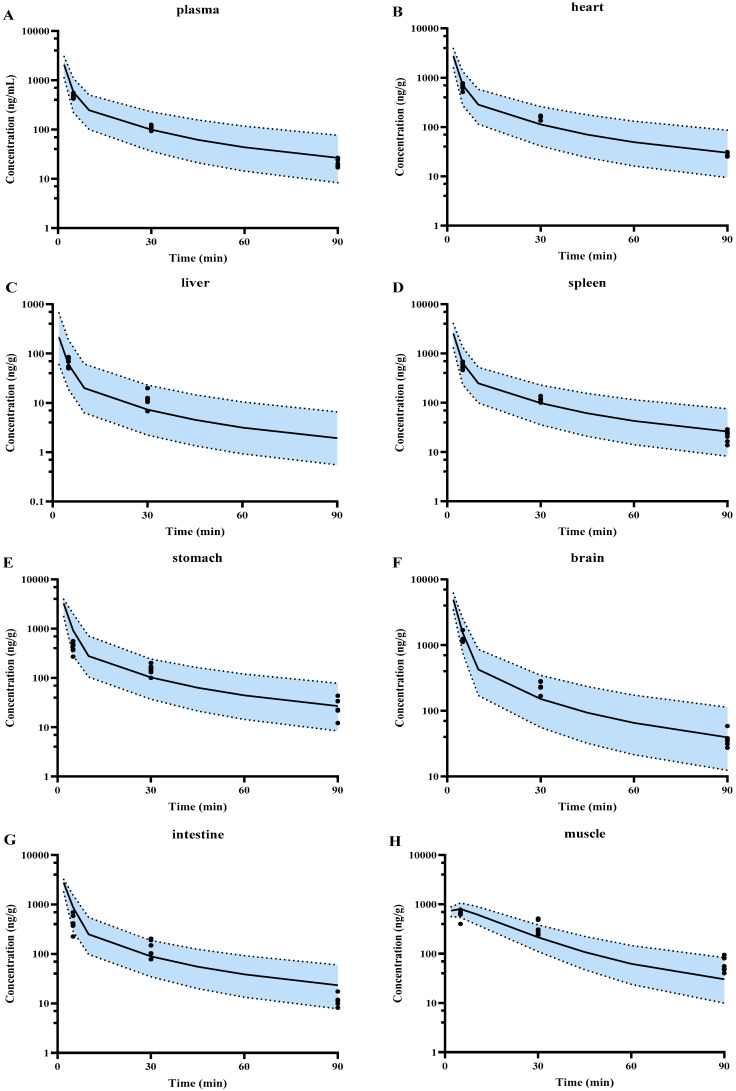
The predicted (lines) and observed (point) concentration-time profiles of SPT-07A in plasma (**A**) and tissues (**B**–**K**) of rats, in heart (**B**), liver (**C**), spleen (**D**), stomach (**E**), brain (**F**), intestine (**G**), muscle (**H**), lung (**I**), adipose (**J**), and kidney (**K**) of rats following iv 2 mg/kg. Blue areas represent 90% prediction intervals (5th–95th percentile boundaries; lower and upper black dashed lines, respectively) for a virtual 100 subjects. (**L**) Represents the relationship of observed and predicted AUC_0–90 min_ of SPT-07A in the tissues above, in which solid and dashed lines indicate unity and twofold errors between predicted and observed data, respectively.

**Figure 5 pharmaceutics-16-01596-f005:**
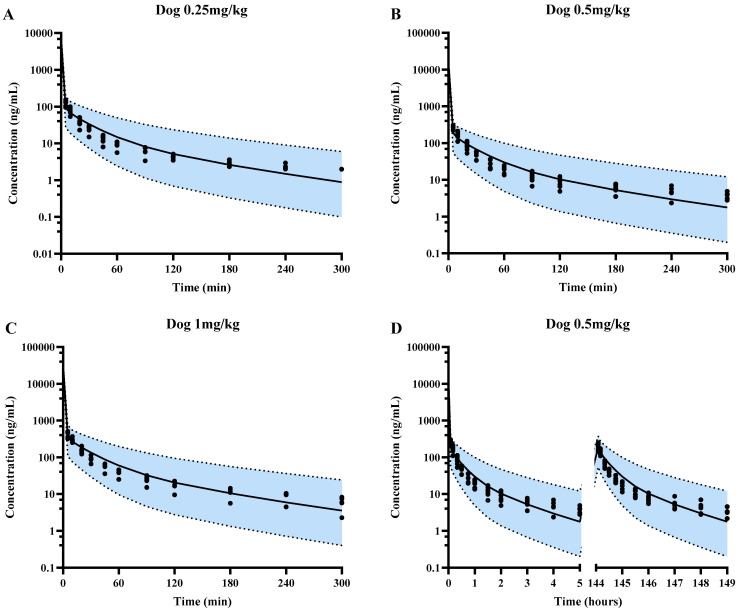
Predicted population mean (black solid lines) and observed (black solid circles) plasma concentration-time profiles of SPT-07A for dogs single-dose 0.25 mg/kg (**A**), single-dose 0.5 mg/kg (**B**), single-dose 1 mg/kg (**C**) and multiple-dose 0.5 mg/kg (**D**). Blue areas represent 90% prediction intervals (5th–95th percentile boundaries; lower and upper black dashed lines, respectively) for a virtual population (n = 100).

**Figure 6 pharmaceutics-16-01596-f006:**
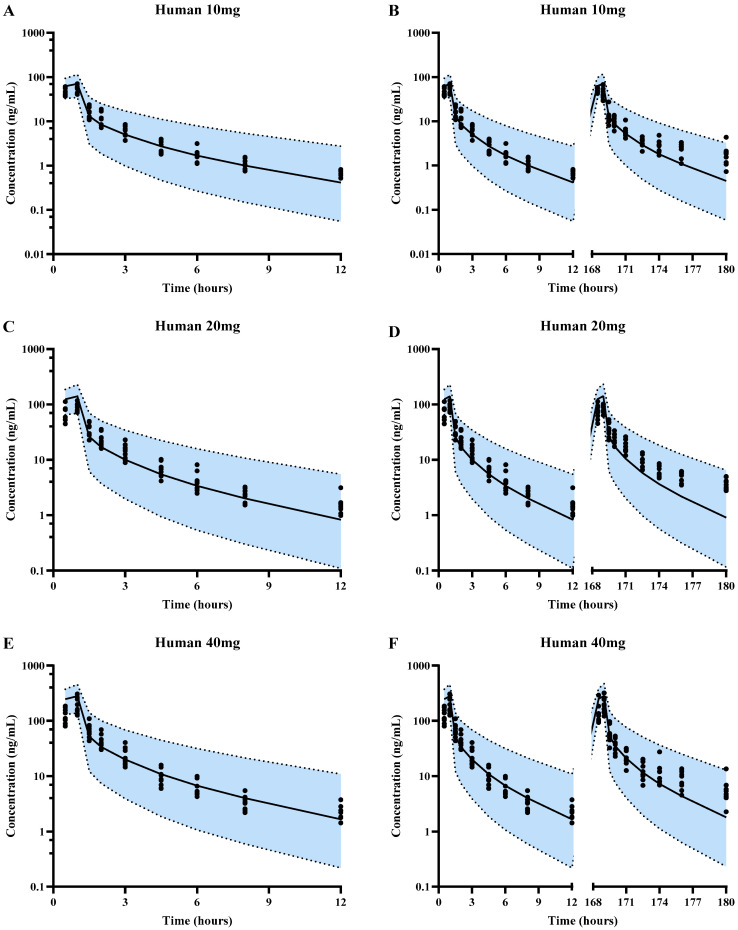
Predicted population mean (black solid lines) and observed (black solid circles) plasma concentration-time profiles of SPT-07A in humans for single-dose 10 mg (**A**), multiple-dose 10 mg (**B**), single-dose 20 mg (**C**), multiple-dose 20 mg (**D**), single-dose 40 mg (**E**) and multiple-dose 40 mg (**F**). Blue areas represent 90% prediction intervals (5th–95th percentile boundaries; lower and upper black dashed lines, respectively) for a virtual population (n = 100).

**Figure 7 pharmaceutics-16-01596-f007:**
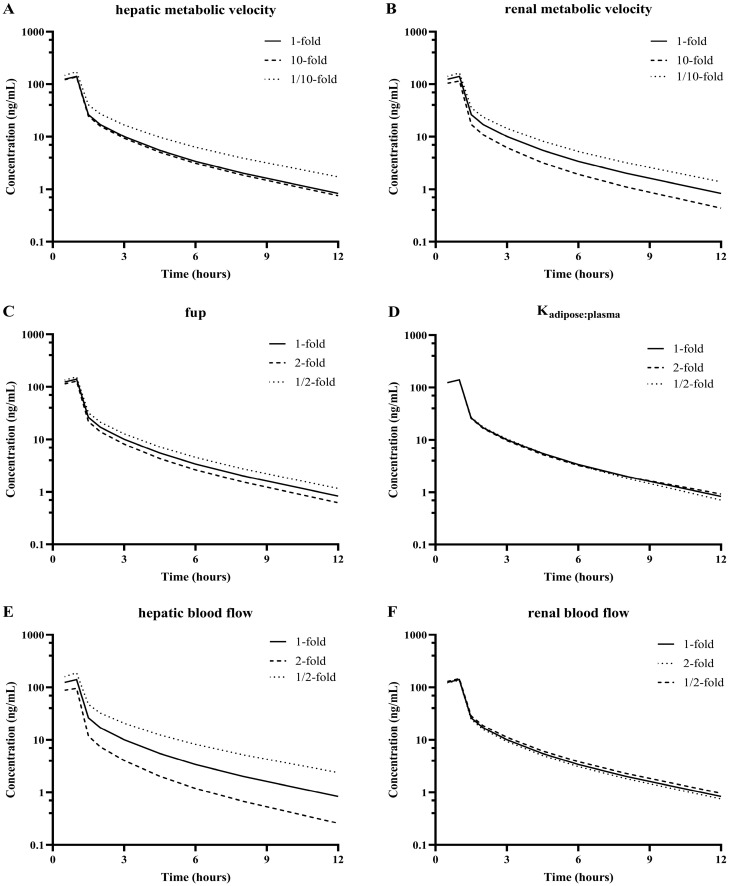
The effects of changing hepatic metabolic velocity (**A**), renal metabolic velocity (**B**), unbound fraction in plasma $f_{u,p}$ (**C**), Adipose to plasma partition coefficients $K_{adipose:plasma}$ (**D**), hepatic blood flow rate (**E**) and renal blood flow rate (**F**) in the human PBPK model on predicted human plasma concentration-time profile of SPT-07A.

**Figure 8 pharmaceutics-16-01596-f008:**
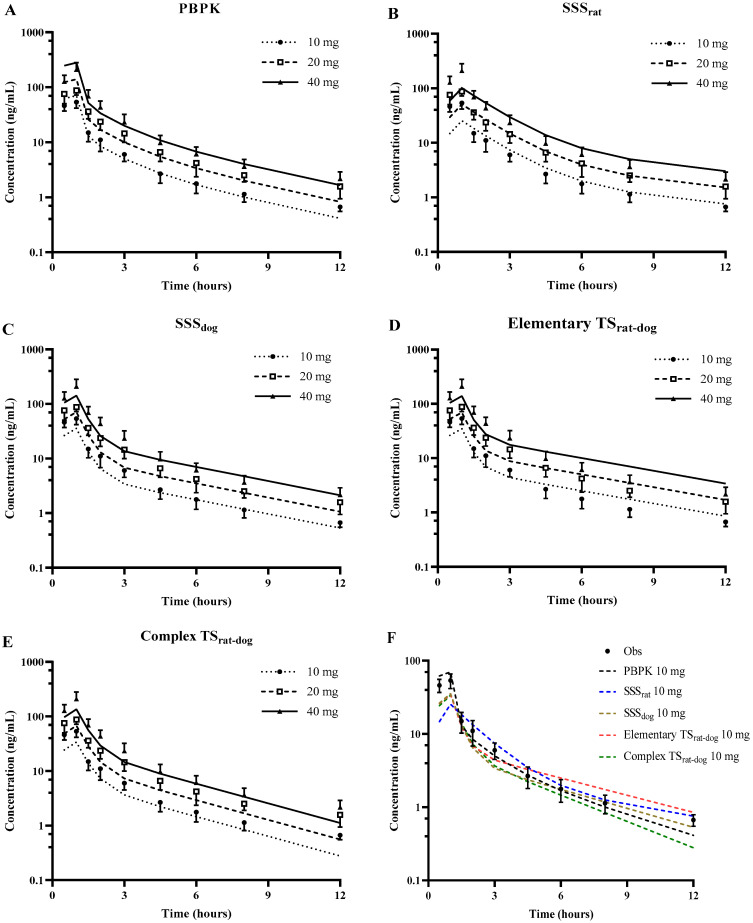
Observed and predicted plasma concentration-time profiles of SPT-07A in humans by the PBPK (**A**), SSS_rat_ (**B**), SSS_dog_ (**C**), Elementary TS_rat-dog_ (**D**), and Complex TS_rat-dog_ (**E**). Compares the human plasma concentration-time profiles of SPT-07A predicted by five methods (**F**).

**Table 3 pharmaceutics-16-01596-t003:** The pharmacokinetic parameters of SPT-07A in rats, dogs, and humans following intravenous administration.

	Dose (mg/kg) ^a^	C_max_ (ng/mL)		C_max_ R ^c^	AUC (μg·min/mL)		AUC R ^d^	CL (mL/min/kg)		CL R ^e^	t_1/2_ (min)		t_1/2_ R ^f^
		Obs	Pre		Obs	Pre		Obs	Pre		Obs	Pre	
Rats	0.5				3.09	3.78	1.22	146	129	0.88	107	58.2	0.54
	1				6.25	7.56	1.21	149		0.87	98.0		0.59
	2				16.0	15.1	0.95	122		1.06	60.2		0.97
	1.0 ^b^				6.82	7.46	1.09	135	131	0.97	97.5	58.0	0.60
Dogs	0.25				3.31 ± 0.717	3.60	1.09	71.7 ± 21.0	67.7	0.94	113 ± 51.2	76.7	0.68
	0.5				7.50 ± 1.18	7.19	0.96	63.1 ± 10.4		1.07	119 ± 12.2		0.64
	1				13.3 ± 2.14	14.4	1.08	71.2 ± 14.9		0.95	118 ± 40.8		0.65
	0.5 ^b^				6.06 ± 1.03	7.19	1.19	77.4 ± 14.3	67.7	0.87	126 ± 38.2	76.7	0.61
Humans	10	55.1 ± 11.6	70.1	1.27	5.06 ± 0.506	5.76	1.14	30.7 ± 3.17	24.3	0.79	231 ± 99.8	179	0.77
	20	90.5 ± 15.7	140	1.55	9.81 ± 1.80	11.5	1.17	33.8 ± 5.61		0.72	251 ± 73.7		0.71
	40	210 ± 71.4	281	1.34	18.3 ± 4.38	23.0	1.26	35.1 ± 12.0		0.69	219 ± 67.9		0.82
	10 ^b^	53.1 ± 9.21	70.5	1.33	5.38 ± 0.895	5.87	1.09	26.5 ± 5.67	23.9	0.90	348 ± 147	180	0.52
	20 ^b^	92.7 ± 12.6	141	1.52	11.4 ± 1.82	11.7	1.03	25.9 ± 4.18		0.92	384 ± 172		0.47
	40 ^b^	203 ± 61.6	282	1.39	20.3 ± 5.11	23.5	1.16	28.0 ± 7.59		0.85	350 ± 119		0.51

^a^ for humans the unit of dose is mg; ^b^ for administration is multiple-dose. ^c^, ^d^, ^e^, and ^f^ was the predicted/observed ratio of C_max_, AUC, CL, and t_1/2_, respectively.

**Table 4 pharmaceutics-16-01596-t004:** The pharmacokinetic parameters ratios for sensitivity analysis.

	Fold Error (Change/Control)			
	C_max_	AUC_last_	CL	t_1/2_
Control 1-fold	/	/	/	/
Liver metabolic velocity 10-fold	0.97	0.96	1.04	0.99
Liver metabolic velocity 1/10-fold	1.22	1.36	0.73	1.08
Kidney metabolic velocity 10-fold	0.81	0.76	1.33	0.95
Kidney metabolic velocity 1/10-fold	1.16	1.24	0.80	1.05
f_up_ 2-fold	0.91	0.88	1.14	0.98
f_up_ 1/2-fold	1.10	1.15	0.86	1.03
K_adipose:plasma_ 2-fold	1.00	0.99	1.00	1.12
K_adipose:plasma_ ½-fold	1.00	1.00	1.00	0.91
hepatic blood flow 2-fold	0.68	0.60	1.68	0.92
hepatic blood flow 1/2-fold	1.34	1.56	0.63	1.14
kidneys blood flow 2-fold	0.97	0.95	1.05	0.99
kidneys blood flow 1/2-fold	1.04	1.06	0.94	1.01

**Table 5 pharmaceutics-16-01596-t005:** The comparison of predicted accuracies in humans by Dedrick and PBPK method.

Method	mg	Fold Error (Predicted/Observed)				GMFE	RMSE
		C_max_	AUC	CL	t_1/2_		
PBPK_human_	10	1.27	1.11	0.89	0.77	0.94	0.17
	20	1.55	1.13	0.86	0.71		
	40	1.34	1.24	0.77	0.82		
SSS_rat_	10	0.46	0.76	1.30	1.16	0.97	0.22
	20	0.56	0.78	1.25	1.06		
	40	0.48	0.85	1.12	1.22		
SSS_dog_	10	0.64	0.68	1.45	0.90	0.79	0.19
	20	0.78	0.70	1.39	0.83		
	40	0.68	0.76	1.25	0.95		
Elementary TS_rat-dog_	10	0.63	0.77	1.28	1.00	0.97	0.20
	20	0.77	0.79	1.23	0.92		
	40	0.66	0.86	1.10	1.05		
Complex TS_rat-dog_	10	0.61	0.64	1.56	0.65	0.71	0.22
	20	0.75	0.65	1.50	0.60		
	40	0.64	0.71	1.34	0.69		

The parameters β_1_ and β_2_ for SSS_rat_, SSS_dog_ were 1.00 and 0.75, respectively; Elementary TS_rat-dog_, where the exponent for β_1_ was 1.00 and β_2_ was derived from interspecies scaling; Complex TS_rat-dog_, where the exponents for both β_1_ and β_2_ were obtained from interspecies scaling.

## Data Availability

Data are contained within the article and the cited clinical trial data were already published by Weicong Wang et al. [7] (NCT01904318).

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
