# Peer review of "Prediction of SPT-07A Pharmacokinetics in Rats, Dogs, and Humans Using a Physiologically-Based Pharmacokinetic Model and In Vitro Data"

_pharmaceutics, 2024, doi:10.3390/pharmaceutics16121596_

Round 1

Reviewer 1 Report

Comments and Suggestions for Authors

The paper has good potential and is a promising exploration topic, however, several aspects need to be addressed. Among them is the english language. 

The paper also is very long as it includes lots steps, if cutting some details is not possible then dividing the paper into two papers may be considered. 

The abstract:

The abstract can use some rearrangement as after mentioning some in vitro results, it starts addressing the simulation method then jumps to results. Also adding a conclusion is needed in the abstract

What is meant by

“SPT-07A is being currently developed in China for treatment of ischemic stroke, but information of its pharmacokinetics is limited” what is the missing information, is the information limited or the details of the PK are limited as for any drug to be submitted for approval the drug PK should be available for review by.

I am not sure if this statement is correct
“Modeling can help replace animal trials”

The method

Line 85: why are there concentrations picked why not a multiples of 10

What is the total number of rats. This should be indicated in section 2.3

What is the software used, more details are needed,

A table with all the parameters used for the model is needed

Table 2 and figure 2 are missing standard deviation or errors of all these values

Figure 7 resolution is low

The following are some typos but please check the whole paper

 There are frequent missing “%” throughout the paper

There are multiple flipped phrases like “ intravenously received single dose” instead of “single IV dose”

intravenous multiple-dose rather than intravenous multiple IV-dose

172 all titles should be consistent

. Tissue distribution in Rats: capitalize D

Line 238: is missing the details of respectively

What is meant by metabolic velocity? In line 264?

Line 275 is defining D, while there is no D in the equation

Results in table 1, in dogs DLM:  UGT and CYP,  CLint,in vivo,u and the Total CLint,in are the same, 25500. Please check calculations

Please indicate the ratio of Observed/predicted in the table 3

Line 442 should read flow not hepatic blood follow

The following does not read 437 to 440

Check the use of the words obviously and little

What the ratio always between 0.5 to 2 for all PK parameters at all dosing levels

Is there any suggested explanation for: SPT-07A in rat  sex differences

PBPK modeling is already in use and thus the final statement of . We hope that this case will inspire scientists to develop drugs for other global health diseases using PBPK modeling at drug discovery and development phase. Is not needed

More discussion of the results is needed and a good explanation on the novelity of the result in the potential application +

More details on the software is needed

Comments on the Quality of English Language

The vocabulary, punctuation, and Grammar of the paper need review. I have indicated very few examples but have not able to spend more time correcting it.

There are also some words in italics and some that need to be in italics but are not (eg in vitro)

Many sentences are missing articles (A, Am, The)

Has a few typos like missing the % in few results. Also “respectively” should come at the end of the sentence.

Line 57: The aim should be the aims

Line 105: extra “or”

Line 143 has extra in

I am not aware of what is meant by “blood clearance” its not a commonly used term, Is this the same as Clearance or body clearance?

Line 341, systematic should read systemic clearance

Line 378 is missing respectively 

Reviewer 2 Report

Comments and Suggestions for Authors

The article is important, but first, it should explain the hypothesis better. Why was PBPK chosen, and how could the finding become a new commercial drug?

The authors should revise their work for improvement and clarification of the points above:
- PBPK Model Details: While the report outlines the development of the PBPK model, a more comprehensive rationale for selecting this model might enhance the study. What renders PBPK the most appropriate method for predicting the pharmacokinetics of SPT-07A in comparison to alternative approaches?
- In Vitro Metrics - The use of liver, kidney, and intestine microsomes from diverse species is a commendable selection; nonetheless, the research should elaborate on the rationale for the particular choice of these tissues and address any limitations associated with employing these models to forecast the drug's action in humans.
- While the article establishes a commendable link between in vitro and in vivo data, it is crucial to underscore the constraints of this extrapolation. The study may elaborate on the disparities between experimental circumstances and actual clinical scenarios, including individual enzyme variations (e.g., genetic diversity in UGT expression) and the impact of variables such as age, food, or health state.

-Variations in Metabolizing Enzymes: The research indicates that UGT2B7 is pivotal in the metabolism of SPT-07A; nevertheless, a more comprehensive discussion of the ramifications of this discovery would be valuable.

-The sensitivity analysis conducted in the paper is commendable, however it may benefit from further elaboration. What are the essential characteristics influencing the pharmacokinetics of SPT-07A, and how might minor alterations in these parameters impact the drug's behavior?

- Limitations of In Silico Models: While the PBPK model is a robust methodology, it is not without constraints, similar to any model reliant on in vitro data and simulations. It would be beneficial to discuss how the model may be enhanced or modified to incorporate new data or unexamined circumstances, such as interactions with other drugs.

Also: the total amount of rats should be addressed and why the animals were not cannulated, reducing animal use?

Round 2

Reviewer 2 Report

Comments and Suggestions for Authors

no comments

Comments on the Quality of English Language

should be revised